# Influence of ROCK Pathway Manipulation on the Actin Cytoskeleton Height

**DOI:** 10.3390/cells11030430

**Published:** 2022-01-26

**Authors:** Carolin Grandy, Fabian Port, Jonas Pfeil, Kay-Eberhard Gottschalk

**Affiliations:** Institute of Experimental Physics, University Ulm, 89081 Ulm, Baden-Württemberg, Germany; carolin.grandy@uni-ulm.de (C.G.); fabian.port@uni-ulm.de (F.P.); jonas.pfeil@uni-ulm.de (J.P.)

**Keywords:** actin cytoskeleton, ROCK pathway, metal-induced energy transfer, z-dimension

## Abstract

The actin cytoskeleton with its dynamic properties serves as the driving force for the movement and division of cells and gives the cell shape and structure. Disorders in the actin cytoskeleton occur in many diseases. Deeper understanding of its regulation is essential in order to better understand these biochemical processes. In our study, we use metal-induced energy transfer (MIET) as a tool to quantitatively examine the rarely considered third dimension of the actin cytoskeleton with nanometer accuracy. In particular, we investigate the influence of different drugs acting on the ROCK pathway on the three-dimensional actin organization. We find that cells treated with inhibitors have a lower actin height to the substrate while treatment with a stimulator for the ROCK pathway increases the actin height to the substrate, while the height of the membrane remains unchanged. This reveals the precise tuning of adhesion and cytoskeleton tension, which leads to a rich three-dimensional structural behaviour of the actin cytoskeleton. This finetuning is differentially affected by either inhibition or stimulation. The high axial resolution shows the importance of the precise finetuning of the actin cytoskeleton and the disturbed regulation of the ROCK pathway has a significant impact on the actin behavior in the z dimension.

## 1. Introduction

The cells cytoskeleton gives mechanical support, allows the cell to deliver cargo and is fundamental for cell division. It consists of three biological polymers: microtubules, intermediate filaments and actin. The actin cytoskeleton is predominantly responsible for cellular movement. It is tightly regulated and highly dynamic. Actin is located in a variety of areas of the cell (Figure 1a). It is found in the cell cortex to give the cell strength and shape [1]. To divide cells, it also forms the contractile ring between the nuclei [2,3]. For motility, it forms lamellipodia and filopodia [4,5,6,7].

The actin cytoskeleton can create contractile forces in the cell. Contractile forces are generated when actin filaments recruit myosin II motor proteins [7,8]. The poly-protein complex of contractile bundles of actin filaments with myosin II molecules are called stress fibers [9]. In contrast, protrusive forces arise from the coordinated polymerisation of multiple actin filaments. Gradients of tension induce local contractions and force cell movements and deformations. [7,8]. These forces are not only important in cell migration, but also play a role in determining cell shape, adapting to the mechanical properties of the environment, determining intracellular movement [10] and morphogenesis of membrane organelles [11]. Filamentous actin (f-actin) is also involved in the generation of mechanical forces which stimulate the differentiation and development of stem cells [12].

Regulated polymerization and depolymerization of actin monomers to polymers leads to a dynamic assembly and disassembly of actin fibers, which ensures quick cellular adaptation to changes in environment [10,13].The small Rho GTPase RhoA is a major regulator of the cytoskeleton (Figure 1b). It activates the Rho-associated coiled coil-containing protein kinase (ROCK) [14]. Through a positive feedback loop, ROCK directly catalyses myosin light chain phosphorylation and inhibits myosin light chain phosphatase [15]. This increases the binding of myosin II to actin and thus the contractility of the cell. ROCK also phosphorylates LIM kinase (LIMK), which then phosphorylates cofilin [16]. This deactivates actin depolymerization and the existing actin filaments are stabilized and increase in length.

High expression of ROCK/LIMK/Cofilin and other members of the ROCK pathway have been identified in cancers such as breast, prostate, colorectal and bladder [17,18,19,20]. In tumour cells, increased movement, invasion and metastasis also occur due to the actin cytoskeleton and its adaptation [7]. Viruses modify cells and the actin to best suit their hosts and proliferate [21,22]. Cell movement through the actin cytoskeleton plays a critical role in many pathologies. During chronic inflammatory diseases migration of immune cells is important [23,24] and actin dynamics are a target for therapeutics against chronic kidney diseases [25]. Ageing processes also change the actin expression and dynamics [26]. Therefore, actin modifications are involved in ageing, cancer, vascular diseases, and neurogenerative diseases like Alzheimer’s disease [26,27]. Disrupted functioning of actin is also observed in somatic cells, stem cells and gametes [12,26]. This underlines the necessity to gain a deep understanding of the Rock pathway on actin structure. In the past, a variety of high resolution fluorescence studies described the actin organization with high precision in two dimensions [28,29,30]. However, the third dimension, perpendicular to the focal plane, is more difficult to resolve with nanometer resolution. Metal induced energy transfer is a tool that facilitates a high precision analysis of this third dimension.

With metal induced energy transfer (MIET) as a tool, the position of fluorescent molecules above a metal surface can be determined with nanometer precision [31,32,33,34]. MIET has an axial resolution limit up to 3 nm [31]. The principle of MIET compares best to Förster Resonance Energy Transfer (FRET). In FRET the energy of an excited donor dye is transferred to a second dye, the acceptor [35,36]. However, at MIET, the acceptor molecule is replaced by a metal layer. This results in a distance-dependent energy transfer rate between the metal layer and the donor molecules, which can directly be correlated to the fluorescence lifetime [31] and is observable at distances from the metal of more than 100 nm. The effect is based on strong optical near-field coupling of surface plasmons in a thin metal layer [32]. The fluorescence lifetime increases non-linearly with increasing distance to the metal layer and asymptotically reaches the lifetime of a free fluorophore without metal layer [31,33]. The fluorescence lifetime can hence be converted to a distance value using a calibration curve to obtain super-resolution in the Z dimension [31]. MIET imaging has already been used to study the cell mechanics of human mesenchymal stem cells [33], to better understand human blood platelet spreading [37] and for the examination of microtubules [38] and the nuclear envelope [39].

In this study, we investigate the regulation of the actin cytoskeleton through different targeting sites in the ROCK pathway. To this end, we use Blebbistatin as a myosin II inhibitor [40], Y27632 as an inhibitor for ROCK [41,42] and the Rho Activator II, which increases the activity of RhoA [43,44]. We use MIET to quantitatively observe the basal height of actin stress fibers and the perimeter of the actin cytoskeleton and compare it to the height of the basal membrane as control. The actin perimeter consists mostly of cortical actin. Hence, this perimeter reflects partially parts of the actin cortex. MIET has a much better resolution than an optical microscope, thus it is possible to detect differences in height within nanometer accuracy. With this, we gain a better understanding of the structure of the actin cytoskeleton in the third dimension and thereby the influence of the different key points of the ROCK Pathway and their impacts on the three-dimensional actin structure. Figure 1a shows a sketch of an adherent cell with its actin cytoskeleton (red). In this work, we analyze both stress fibers, which are formed with the aid of myosin II, and the actin edge (blue), which is composed of various actin components such as cortical actin as well as stress fibers. In addition, the gradient of actin height along the actin edge is analyzed. The actin structures in the area of the cortex are of growing interest in the scientific community [45,46,47]. Although they are not yet well studied in interaction with the ROCK pathway, the cortex plays an important role as it adapts and changes the cell shape in a highly sensitive manner [45,47].

Our results show that by treatment with Rho Activator, actin distance to the substrate is higher and distributed on a smaller projected cell area. In contrast to this, the treatment with the inhibitor Y27632 leads to lower actin heights. Blebbistatin, which only affects myosin II activity but does not directly interfere with the actin assembly pathway, does not lead to major changes in height, but flattens the actin edge of the cell. Our results indicate that there is a precise balance between adhesion tension and cytoskeleton tension of the actin cytoskeleton regulating the three-dimensional architecture. A disturbance of this balance disrupts the finely balanced interplay of forces and leads to flattened actin architecture.

## 2. Materials and Methods

### 2.1. Coating for MIET Measurements

For MIET measurements glass coverslips (Paul-Marienfeld, Lauda-Königshofen, Germany) were silanized by evaporation with (3-Mercaptopropyl)trimethoxysilane (MPTMS) (Sigma-Aldrich, Darmstadt, Germany) and afterwards coated with 20 nm gold (Kurt-Lesker) as previously described [48]. To apply fibronectin to the gold surfaces, the gold slides were treated with the crosslinker Dithiobis (succinimidyl propionate) (DSP) (Thermo-Fisher, Waltham, MA, USA) in Dimethyl sulfoxide (DMSO) (4 mg/mL) for 30 min. Afterwards coverslips were washed in Phosphate buffered saline (PBS) and coated with 0.025 mg/mL fibronectin (Sigma-Aldrich) for 2 h. Fibronectin as an extracellular matrix protein is used to bind fibroblasts to integrins via its RGD sequence.

### 2.2. Cell Culture, Drug Treatment and Immunofluorescence Staining

NIH 3T3 mice fibroblasts (ATCC^®^ CRL-1658™, Manassas, VA, USA) were cultured in Dulbecco’s Modified Eagle Medium (Gibco by Life Technologies, Waltham, MA, USA) containing 10% Fetal Bovine serum (Bio&Sell, Feucht, Germany) and 1% antibiotic/antimycotic solution (GE Healthcare Hyclone, Chicago, IL, USA). For MIET measurements cells were trypsinized with Trypsin/EDTA (Biowest, Nuaillé, France) for 3 min and seeded on fibronectin coated gold coverslips for 24 h in the incubator.

The cells were treated with different drugs the next day. Drug treatment was 50 µM Blebbistatin (Sigma-Aldrich, Darmstadt, Germany) for 40 min, 10 µM Y-27632 (Sigma-Aldrich, Darmstadt, Germany) for 1 h and 1 µg/µL Rho-Activator II (Cytoskeleton, Denver, CO, USA) for 3 h, respectively. After each drug treatment cells were briefly rinsed 3× in PBS, which was immediately removed and then fixed in 4% formaldehyde (Polysciences, Warrington, PA, USA) for 10 min. The addition of drugs was always performed at the same time of cultivation. For cell membrane staining, cells were incubated 5 min in 5 µg/mL Cell Mask Orange (Thermo-Fisher, Waltham, MA, USA) before fixation.

For immunofluorescence staining cells were permeabilized in 0.8% Triton X-100 (Sigma-Aldrich, Darmstadt, Germany). For blocking they were incubated in 0.1 M Glycin (Alfa Aesar, Haverhill, MA, USA) for 30 min and 3% Bovine serum album (Sigma-Aldrich, St. Louis, MO, USA) for 30 min. The actin cytoskeleton was stained by incubating 66 µM Alexa Fluor 568 phalloidin (Thermo-Fisher, Waltham, MA, USA) overnight. The nucleus was stained with DAPI (AppliChem, Darmstadt, Germany) in a concentration of 0.002 µg/mL for 5 min. Except of the overnight staining at 4 °C all steps of immunofluorescence staining are performed at room temperature. For imaging PBS was used as mounting medium.

### 2.3. Experimental Setup

MIET measurements are performed on a fluorescence lifetime imaging microscopy setup MicroTime 200 of PicoQuant (Berlin, Germany). The setup is attached to an IX73 inverted microscope (Olympus, Hamburg, Germany). Fluorophores are excited via a pulsed diode laser (LDH-D-TA-560, pulse width ~56 ps, repetition rate 40 MHz, wavelength 560 nm), which is coupled into a polarization maintaining single mode fiber. For both focusing excitation light and collecting fluorescence light a high numerical aperture objective (60x1.2 UPlanSApo, Superapochromat, water immersion, WD = 0.28 mm) was used. The collected fluorescence has been splitted by a dichroic beam splitter (zt488/861rpc-UF3, AHF/Chroma, Tübingen, Germany), passing the pinhole and is filtered by an emission short-pass ET750sp-2p8 (AHF/Chroma) and emission long-pass filter BLP01-594R (AHF/Chroma) to the hybrid-PMT detector. For data acquisition the multichannel picosecond event timer HydraHarp 400 TCSPC module of PicoQuant was used. FLIM images of each single cell were recorded with the SymPhoTime 64 software (PicoQuant).

### 2.4. Fluorescence Lifetime Data Evaluation

Fluorescence photons were detected in time-tagged, time-resolved mode, which makes it possible to collect all photons of single pixels and sort them into a histogram according to their arrival time after the last laser pulse. A multi-exponential deconvolution fit with a self-recorded instrument response function was applied to the data sets in the SymPhoTime 64 software (PicoQuant, Berlin, Germany).

### 2.5. Converting Lifetime to Height Values

The MIET GUI programme of the Enderlein group (University of Göttingen) in MATLAB was used to generate the calibration curve to get the height data from the lifetime data. The theory of how to generate the height data from the lifetimes was described in detail previously [31,32,34,49,50]. The layers of the MIET samples were all of the same composition. On the bottom of the sample was glass (refractive index *n* = 1.52) and a layer of MPTMS (*n* = 1443 [51]), which was coated with 20 nm of gold. The refractive index of gold is wavelength-dependent and was taken from [52]. Above the gold is the cell (*n* = 1.36 [53]) and PBS (*n* = 1.33 [54]) as mounting medium. The calibration curve was then used to calculate the height values in a custom-written MATLAB code, which was adapted from the MIET GUI [31,34].

### 2.6. MATLAB Height Analysis

For further analysis of the calculated heights the MATLAB ver. R2019b software (The MathWorks, Inc., Natick, MA, USA) was used. In order to determine the edge of the cell, the image was binarized and the largest object was selected. The edge of the largest object corresponds to the actin edge and was calculated with pixel accuracy. To avoid binarization artefacts, the cell was eroded by 2 pixels. Areas of cells larger than the image section were not included in the edge analysis. For the determination of the gradient angle along edge, the cell height along the edge line was derived and the angle was calculated for every edge pixel accordingly. To normalize the delta of the 90th percentile to the 10th percentile of stress fiber height, a parameter we call relative stress fiber rise, the difference was divided by the 90th percentile.

### 2.7. Extract Stress Fibers

To extract the stress fibers from the intensity weighted height images, these images were analyzed using the TWOMBLI plugin [55] in Fiji [56,57]. The resulting masks were applied to the existing datasets in MATLAB.

### 2.8. Quantification and Statistical Analysis

The populations used for the analysis of the entire study are composed of the following number of single cells: Untreated *n*= 39, Blebbistatin *n* = 38, Y27632 *n* = 39, Rho Activator *n* = 48. Plots in Figures 2, 3 and 6 are generated using MATLAB. Boxplots are generated using PlotsOfData [58]. For statistical analysis Kruskal-Wallis test with post hoc Dunn’s test was performed in R. ns: *p* > 0.05, *: *p* ≤ 0.05, **: *p* ≤ 0.01, ***: *p* ≤ 0.001, ****: *p* ≤ 0.0001.

## 3. Results

The major goal of this study was the examination of the distance from the substrate to the actin cytoskeleton, a parameter we call “actin height”. We analyzed the height of all basal actin components together; the height of actin stress fibers; and the height of the actin edge—the outer edge of the actin in the cell—separately (Figure 2). Further, we analyzed the gradient of the actin height along the edge of the cell, a parameter that shows us the roughness of the cell edge. As control, we measured the height of the basal cell membrane. Figure 2 shows representative images of the actin cytoskeleton generated via fluorescence lifetime imaging microscopy (FLIM). Using a previously created calibration curve, the fluorescence lifetimes were converted into height values [31,32,34,49,50] to get intensity-weighted height image as described in Methods. In order to also get specific height information about stress fibers, a stress-fiber specific mask was generated unbiased via the TWOMBLI plugin [55]. For further analysis of the actin perimeter, the actin edge was extracted from the height image of all components. To examine the regulation of actin by the pathway, untreated cells (Figure 2a) were treated with different drugs. Blebbistatin inhibits myosin II (Figure 2b), Y27632 inhibits ROCK (Figure 2c) and the Rho activator increases the activity of RhoA (Figure 2d).

### 3.1. Height Distribution of All Basal Actin Components

Actin plays a central role in the cell, hence it is the critical player in various cellular behaviors, therefore it is not surprising that actin behaviour is disturbed in many disease patterns [4,10,20]. For this reason, we first analyzed the level of total actin. From the intensity weighted height images (Figure 2b), the median height of each cell’s actin components was evaluated (Figure 3a). Untreated cells have a median of 114 ± 12 nm. Cells treated with the ROCK inhibitor Y27632 show the lowest median at 97 ± 15 nm, followed by cells treated with the myosin inhibitor blebbistatin at 110 ± 17 nm. No significant difference between blebbistatin treated cells and untreated cells can be found. Significantly, Rho activator treated cells show the highest median at 124 ± 12 nm.

### 3.2. Height Distribution of Stress Fibers

Actin stress fibers are contractile actomyosin bundles that play a central role in cell adhesion, morphogenesis, and mechanics [11,12,24]. The median of the stress fiber heights of each cell (Figure 3b) shows a significant difference of the Rho activator population with a median of 132 ± 10 nm compared to all other populations. This is even higher (+ ≈10 nm) compared to the median of the total actin of the Rho activator population (Figure 3a), indicating a significant upshift of the stress fibers compared to other actin structures. The height of stress fibers of Y27632 treated cells are clearly shifted to lower heights compared to untreated cells. Despite a higher median of the height of actin stress fibers in blebbistatin treated cells, there is no significant difference between blebbistatin and Y27632 treated cells or between blebbistatin-treated and untreated cells. The MIET data reveal that untreated cells (Figure 2 and Figure 4a) have arch-like structures between the adhesion points. The blebbistatin -treated cells (Figure 2 and Figure 4b) show an overall similar architecture. The Y27632 treated cell contains very few stress fibers and has very limited actin overall. The few stress fibers show arch-like structures (Figure 2 and Figure 4c). The Rho activator cell (Figure 2 and Figure 4d) has actin stress fibers at a higher height. The stress fibers, however, show no arch-like features.

We further compared the height of the lowest and highest stress fiber regions. To this end, we extracted the regions corresponding to the 10th percentile and 90th percentiles for each of the cells (Figure 3c–f). This is a measure of the three-dimensional height distribution: highly curved actin structures will display a larger difference between the percentiles than flat actin structures. For untreated cells, the 10th percentile is at 90 ± 10 nm, while the 90th percentile is at 147 ± 11 nm. This leads to a rise of the actin arches of 56 ± 7 nm and a relative rise of 38 ± 4%. Blebbistatin shows no significant difference compared to untreated cells. The Rho activator-treated cells have a significantly higher 10th percentile at 106 ± 12 nm and 157 ± 9 nm for the 90th percentile. Its actin arcs increase with 50 ± 6 nm and have a relative rise of 32 ± 5%. Y27632 treated cells, on the other hand, have significantly lower heights in the 10th percentile at 80 ± 11 nm, but are not different compared to blebbistatin treated and untreated cells in the 90th percentile with 146 ± 15 nm. However, they show differences in the rise of actin arches of 59 ± 7 nm and have a relative rise of 42 ± 5%.

### 3.3. Actin Border Analysis

Cell deformations must be precisely controlled as they are key to cell migration, differentiation, division and tissue morphogenesis. These parameters play a significant role in carcinogenesis [45,59]. Changes in cell shape are driven by tension gradients in the cellular cortex [47]. This consists of a thin actomyosin network formed by cortical actin and stress fibers in addition to actin mesh [45,47,59]. To gain deeper insight into this important outer edge of the cellular cytoskeleton, the height behaviour at the boundary of these actin structures is analyzed. We are investigating not only its height, but also its roughness. Thus, both aspects are combined to study the role in diseases affecting the ROCK pathway. The here analyzed actin perimeter only partially reflects the actin cortex: the cortex is larger than the here described structures, and part of the cell edge may also be formed by stress fibres.

Due to the high axial resolution, Y27632 treated cells (Figure 5a) were discovered to have the lowest height (103 ± 15 nm) of all conditions. In contrast, the perimeter of the actin cytoskeleton of Rho activator-treated cells is shifted to higher heights, while the actin edge of untreated (116 ± 13 nm) and blebbistatin-treated cells (114 ± 18 nm) have a similar median.

In addition to analyzing the height of the actin and the influence of the drugs on it, the derivative of the edge height along the edge (Figure 5b) was also analyzed. For further analysis of the steepness distribution of the actin edge, we analyzed the slope of the edge according to its gradient angle defined by $\vartheta =\arctan\frac{\partial h}{\partial x}$, where dh is the change in height of a lateral distance dx. Blebbistatin (3.0 ± 0.5°) and Y27632 (3.0 ± 0.8°) treated cells show a similar overall median and significantly lower gradient angles compared to untreated and Rho activator-treated cells, respectively. There is no significant difference between untreated and Rho activator-treated cells. However, untreated cells have a broader distribution and higher variation compared to cells treated with the Rho activator.

By correlating the heights of the actin edge with the gradient angle along the edge untreated cells Figure 5c) show the widest distribution of angles. The highest density and thus the most heights occur between 95 and 140 nm with a tangent angle of 3.3 ± 1.1°. Blebbistatin (Figure 5e) and Y27632 (Figure 5f) treated cells show a similar and very broad height distribution especially concentrated in the range of 70 to 130 nm. They occur mainly concentrated at low heights with a tangent angle of 3.0 ± 0.5° (blebbistatin) and 3.0 ± 0.8° (Y27632). Y27632 cells, however, have slightly lower heights compared to blebbistatin treated cells. Blebbistatin cells display two centres of high density. Both populations show similar angles. Lower angles are also more likely to be found at lower heights. The Rho Activator cells (Figure 5d), on the other hand, are clearly shifted to the higher heights. The highest density occurs in the range from 110 to 150 nm. They show more evenly distributed angles than untreated cells.

Table 1 provides a summary (median ± std) of the analysis of the actin cytoskeleton in z dimension using MIET, as well as the gradient angles along the actin edge.

### 3.4. Analysis of Cell Membrane

Additionally to the actin height, the height of the cell membrane was measured under all four conditions (Figure 6a–f as control. The height of the basal cell membrane does not change in response to a manipulation of the ROCK pathway. Thus, it can be excluded that the entire cell is being pulled upwards.

### 3.5. Analysis of Actin in The Focal Plane

Interestingly, the treatment with the respective drugs leads to changes in total cell area. A significantly smaller cell area for the Rho Activator (Figure 7a) cells compared to the other three populations can be seen. The cells treated with Y27632 have a significantly larger actin cell area than untreated and blebbistatin-treated cells. Furthermore, to see how sparse the actin cytoskeleton is and in which ratio the actin is distributed in the cell, the actin area of the fibers obtained using the TWOMBLI plugin was normalised to the cell area (Figure 7b). The Y27632 treated cells have a significantly larger cell area with a low actin content. In contrast the Rho activator-treated cells have a small cell area with a high actin content.

## 4. Discussion

Many diseases like cancer involve the ROCK pathway, which also affect the actin cytoskeleton [17,18,19,20]. Therefore, a deeper understanding of the impact of this pathway on actin organization is of major concern. Previous structural studies predominantly analyzed the actin regulation and the interaction with the ROCK pathway in the two dimensions of the focal plane [60,61]. Mostly due to technical difficulties, the third dimension normal to the focal plane has been examined to a far lesser extent. The three-dimensional structure of actin is important for a thorough understanding of cellular physiology and pathophysiology. For this reason, we analyzed here the third dimension in nanometer resolution using MIET. Previously, Chizhik et al. [33] studied the height distribution of actin stress fibers of human mesenchymal stem cells on gold-coated glass at specific time points. In our study, we examined differentiated embryonic mouse fibroblasts using fibronectin as substrate coating. The average actin height appears to be different in the two studies: while in our study, the 10 percentile lowest structures are in the range of 80 nm–120 nm, the earlier study finds lowest actin heights in the order of 20 nm–40 nm. In contrast to this, the actin heights of embryonic mouse fibroblasts from Kanchanawong et al. [62] compares well to our measured height results. Hence, the height profile may be cell type dependent.

Chizhik et al. focus on the time evolution of stress fibers, while here, we concentrate on the effect of disturbing the ROCK pathway on actin height distributions. To this end, we just used end-point measurements of cells seeded after 24 h. We find profound and novel effects of actin regulatory drugs on the height and architecture of the actin structures. In particular, inhibition at an early point in the ROCK pathway leads to a lowering of actin height, while activation leads to an increase in actin height. These differences in height may be caused by re-structuring of the connection between cytoskeleton and extracellular matrix, so called focal adhesions. Integrins expressed on the cell surface bind to the RGD binding motif on fibronectin and may exert forces, depending on ROCK intervention. Fibronectin has been shown to dynamically alter its fibrous structure depending on external forces, revealing cryptic binding sites [63,64,65]. This may lead to force dependent, integrin-mediated re-organization of the cytoskeleton and to recruitment of further proteins to the focal adhesions [66,67] above a certain force threshold [68]. Recruiting more proteins to re-enforce the connection would lead to higher actin structures in line with our results here. Further studies are necessary to understand this mechanism.

The height resolution offered by MIET allows us not only to determine the height of the structures, but also to calculate inclination angles along actin borders. Chizhik at al. find very low inclination angles of actin stress fibers on the order of 0.15° distinct from our values of around 3°. However, we measure different structures: while the previous paper investigates specific pre-chosen stress fibers, we focus on the edge of actin. This edge is comprised of different actin structures like cortical actin and stress fibers. There is increasing interest in the actin cytoskeleton of the cellular cortex because it cannot be neglected in the propagation of disease [1,46,47]. Changes in cell shape are driven by tension gradients in the cellular cortex [45,47]. Here, it is interesting to observe mainly an effect of inhibitors: these lead to lower angles, while activators hardly change this angle compared to untreated cells.

In order to understand the behaviour of the actin cytoskeleton in the ROCK pathway (Figure 1a) in the focal plane as well as in the z-dimension, we propose a simplified model based on balancing different tension contributions within the cell. Our model relates the balance between cytoskeleton tension and adhesion tension. Cytoskeletal tension pulls the cytoskeleton upwards and hence stretches or remodels the connections between cytoskeleton and basal membrane. Further, since adhesion sites are not altered, the increase in cytoskeletal tension pulls the cell together. Balancing this force is the adhesion tension, which tends to stretch the cell. While similar models have been proposed earlier, this is the first direct experimental observation of the effect of manipulating cytoskeletal tension on the structure of the cytoskeleton with respect to the membrane. Since the actin cytoskeleton is connected to the cell membrane via focal adhesions, the here observed differences in distance between membrane and actin directly reflect remodulation of focal adhesions. Figure 8 shows a possible model for the relationship between adhesion tension and cytoskeleton tension to describe the analyzed height behaviour in relation to the ROCK pathway.

Untreated cells form arch like actin fiber structures with a ruffled actin edge. When blebbistatin is added after formation of the actin stress fibers, only small changes can be observed. However, the ruffled actin edge becomes flattened, showing that the ruffling requires active force generation by the cell, since treatment of the cells with Blebbistatin directly inhibits myosin II activity.

Contrary to blebbistatin, Y27632 does not directly inhibit myosin II, which appears later in the ROCK pathway, but rather inhibits ROCK, which is an important key point in the pathway. By inhibiting ROCK, less myosin light chain is phosphorylated and thus less myosin II is available for stress fiber formation and maintenance of stress fiber tension. In addition, inhibiting ROCK prevents cofilin phosphorylation by LIM kinase. Non-phosphorylated cofilin depolymerizes existing actin and leads to an active degradation of actin. Therefore, there is barely any cytoskeleton tension in the Y27632 treated cells due to the destruction of actin and the lack of stress fibers. The adhesion tension predominates in the previously existing equilibrium. This results in significant wide spreading of the cells and lowering of the actin structures. Interestingly, the actin stress fibers appear to be so weak, that even the reduced myosin II induced tension is sufficient for increased arching of the actin stress fibers. The comparison of blebbistatin and Y27632 clearly shows that although both drugs act similarly, the effect on the actin structure is drastically different.

Cells treated with the Rho activator show an opposite effect to cells treated with inhibitors. The Rho activator increases the activity of RhoA, which in turn increases the activity of ROCK. The myosin light chain is increasingly phosphorylated. In addition, a positive feedback loop inhibits myosin light chain phosphatase. More myosin II is provided for stress fiber formation and maintenance of existing stress fibers. This results in stronger contractions in the cytoskeleton compared to untreated cells. Furthermore, the increased activity of ROCK also causes the LIM kinase to phosphorylate more cofilin. As a result, less actin is depolymerized and existing actin structures are stabilized. Thereby, the actin content is increased to an even greater extent. Due to this stabilization of existing actin structures and the stronger contraction, the cytoskeleton tension increases strongly. This creates an imbalance, causing the actin cytoskeleton to be pulled further up and together leading to rounded shape with a small area and high concentration of actin fibers. The reduced migration behaviour described in the literature [16,69] can also be explained by this imbalance.

The height differences in distance between basal membrane and actin cytoskeleton shown in Figure 6f clearly point to a mechanism, in which the connectors between cell membrane and actin cytoskeleton are being modulated by drugs acting on the cytoskeleton. Our paper shows the ability to further analyse these drug related changes at high resolution.

## 5. Conclusions

As shown in this study, the regulation of the ROCK pathway has an effect not only on the actin cytoskeleton at the focal plane but also perpendicular to it. The ROCK signalling pathway influences various cellular functions by acting on the cytoskeleton. The high axial resolution provided by metal-induced energy transfer (MIET) allows us to precisely measure the height of cytoskeletal structures. However, cell-to-cell variations in shape and size make an even deeper analysis of the effect of these drugs challenging.

In summary, we show that balancing forces in the actin cytoskeleton is required for a proper three-dimensional cytoskeletal architecture. Increasing tension leads to an actin cytoskeleton at large height in cells with a small cross-sectional area, while decreasing cytoskeletal tension also flattens the actin cytoskeleton but close to the cell’s substrate with increased cell area. MIET hence gives unprecedented insight into three-dimensional actin regulation by the ROCK-pathway, enabling the investigation of the fine-tuned balance between adhesion tension, stress fiber formation and force generation. Through this study, more detailed information about dysfunctions in the ROCK pathway with resulting disease patterns due to the effects of the cytoskeleton can be obtained in the future.

## Figures and Tables

**Figure 1 cells-11-00430-f001:**
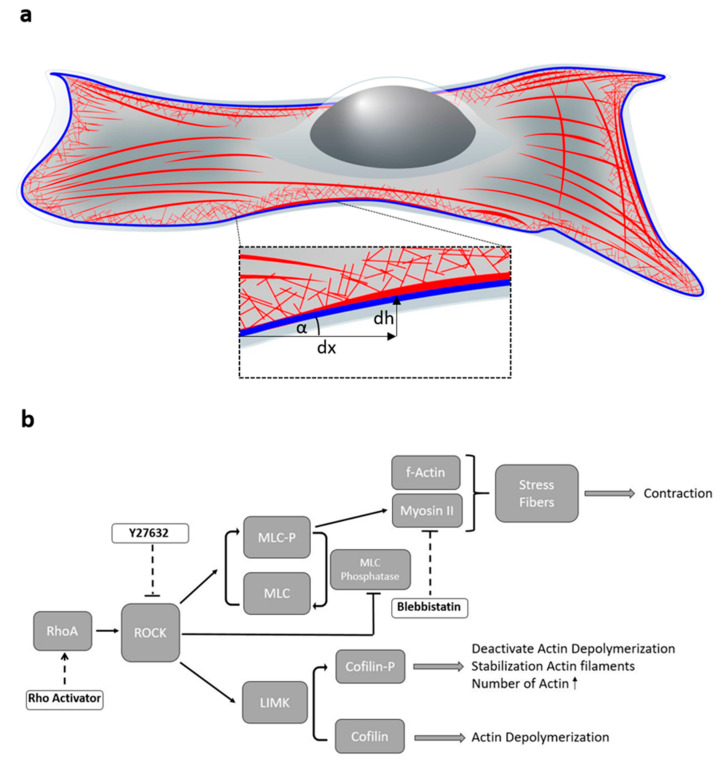
(**a**) Scheme of a cell with actin cytoskeleton (red) and nucleus (grey) shown in 3D. The further analyzed actin edge consists of different types of actin and is shown in blue. The image section shows the parameter angle along the edge, which describes the differentiation of the edge. (**b**) Schematic of the investigated ROCK Pathway and the influence of certain drugs on key points in the pathway. RhoA activates the Rho-associated coiled coil-containing protein kinase (ROCK). ROCK catalyzes the phosphorylation of the myosin light chain (MLC) to myosin light chain phosphate (MLC-P) and inhibits myosin light chain phosphatase (positive feedback loop). This increases the binding of myosin II to actin and thus the contractility of the cell increases. ROCK phosphorylates LIM kinase (LIMK). This phosphorylates cofilin. The existing actin filaments stabilize and increase as actin depolymerization is deactivated. Rho activator enhances the action of RhoA, Y27632 directly inhibits ROCK in the pathway and Blebbistatin inhibits the action of myosin II.

**Figure 2 cells-11-00430-f002:**
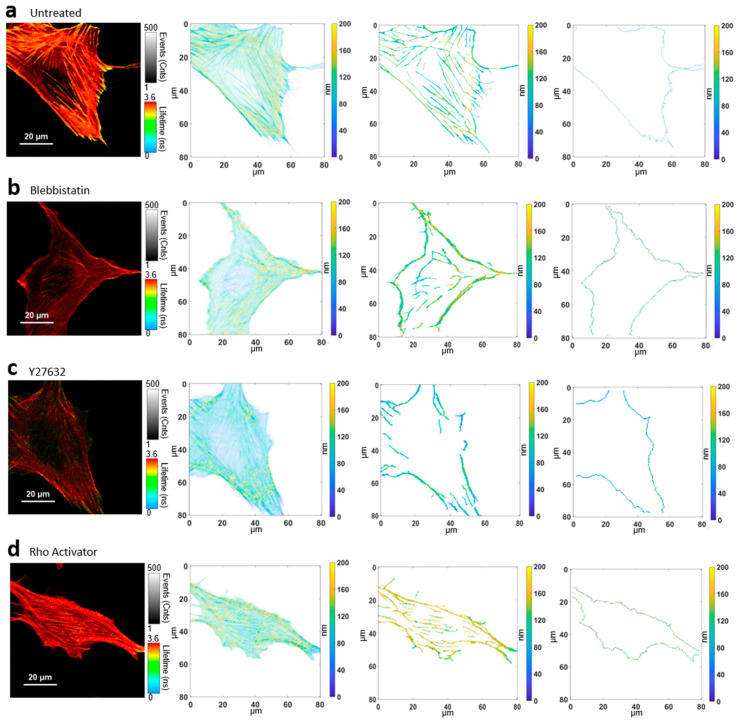
Converting fluorescence lifetime to height values for further analysis. (**a**) Untreated cell: FLIM image, intensity weighted height image of the actin cytoskeleton, extracted stress fiber height image, actin edge image. (**b**) Blebbistatin treated cell: FLIM image, intensity weighted height image of the actin cytoskeleton, extracted stress fiber height image, actin edge image. (**c**) Y27632 treated cell: FLIM image, intensity weighted height image of the actin cytoskeleton, extracted stress fiber height image, actin edge image. (**d**) Rho Activator treated cell: FLIM image, intensity weighted height image of the actin cytoskeleton, extracted stress fiber height image, actin edge image.

**Figure 3 cells-11-00430-f003:**
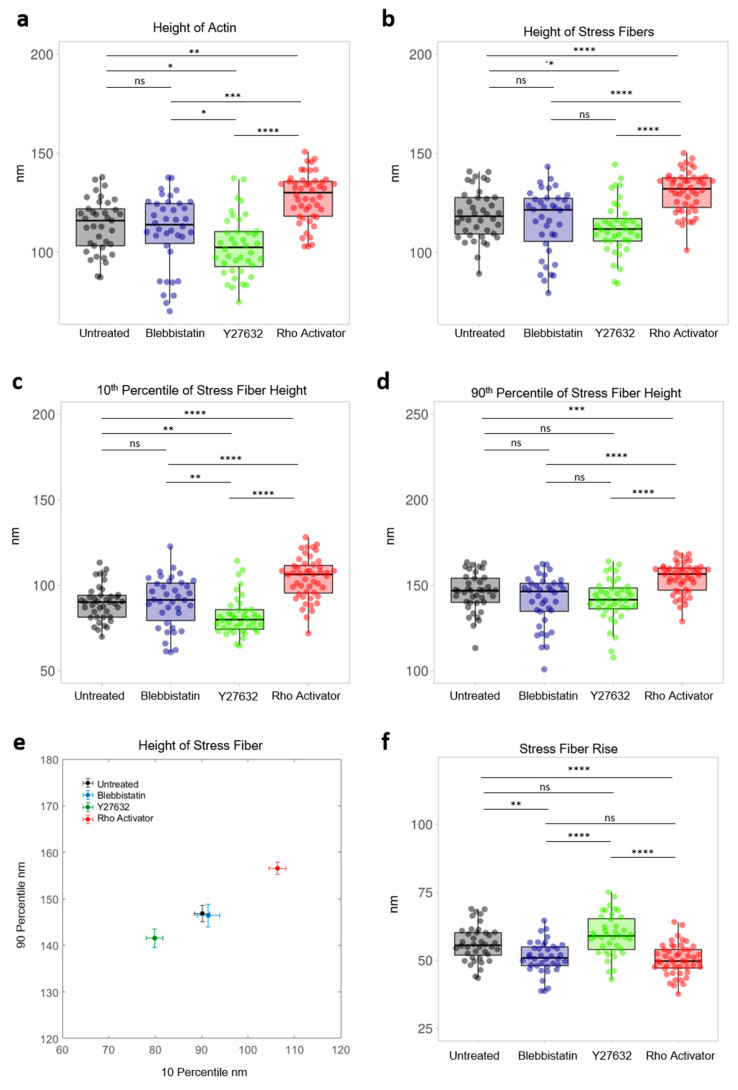
Height analysis of the actin cytoskeleton. Each dot represents the median of a single cell. (**a**) Height of Actin of untreated and blebbistatin, Y27632, Rho aActivator-treated cells. (**b**) Height of stress fibers of untreated cells and cells treated with blebbistatin, Y27632 and Rho activator. (**c**) 10th percentile of stress fiber height of actin of untreated and blebbistatin, Y27632, Rho activator treated cells. (**d**) 90th percentile of stress fiber height of untreated cells and cells treated with Blebbistatin, Y27632 and Rho activator. (**e**) Median of 10th percentile versus median of 90th percentile stress fiber height of untreated and blebbistatin, Y27632, Rho activator-treated cells. (**f**) Relative stress fiber rise is the normalized difference of the 90th to 10th percentile of stress fiber height. Test: Kruskal-Wallis and post hoc Dunn’s test. ns: *p* > 0.05, *: *p* ≤ 0.05, **: *p* ≤ 0.01, ***: *p* ≤ 0.001, ****: *p* ≤ 0.0001.

**Figure 4 cells-11-00430-f004:**
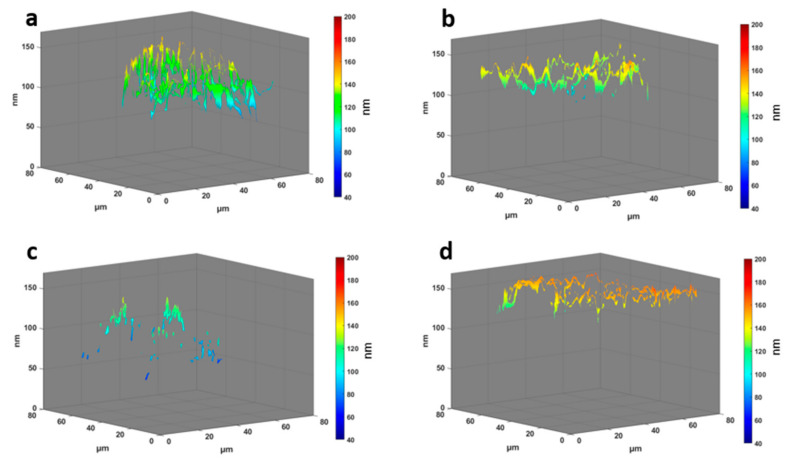
3D plots of the actin distribution with height information in z dimension. The arrow represents the polarization of the cell. The cells shown are the cells from Figure 2 and representative of their population. (**a**) Untreated cells. (**b**) Blebbistatin-treated cells. (**c**) Y27632 treated cells. (**d**) Rho activator-treated cells.

**Figure 5 cells-11-00430-f005:**
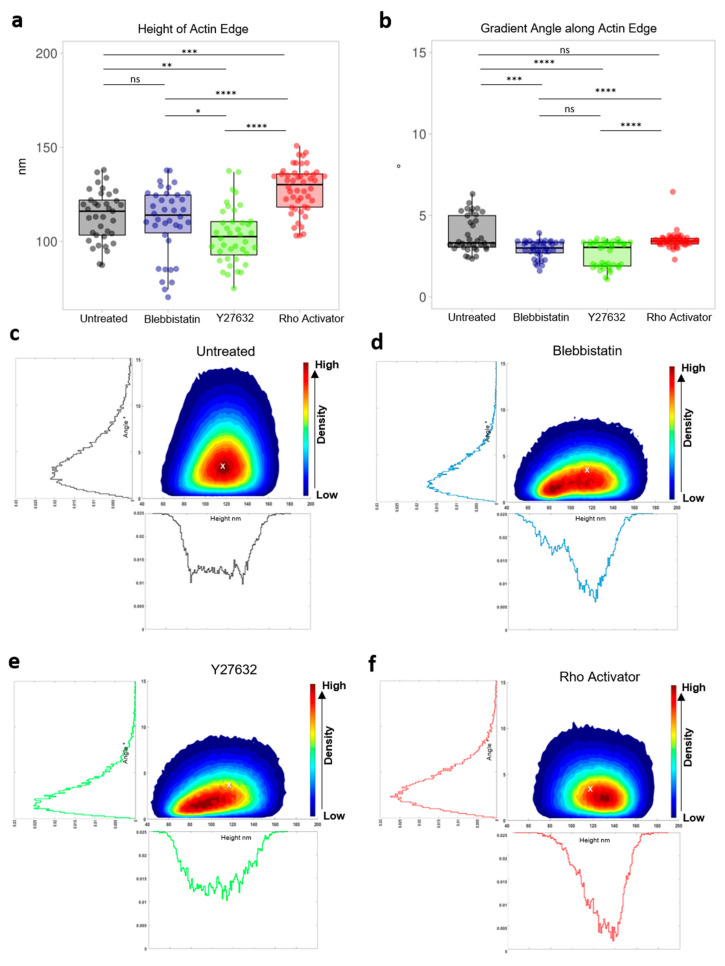
Actin edge analysis. (**a**) Height of actin edge of untreated and blebbistatin, Y27632, Rho Activator treated cells. Each dot represents the median of a single cell. (**b**) Gradient angle along actin edge of untreated cells and cells treated with blebbistatin, Y27632 and Rho activator. Each dot represents the median of a single cell. Test: Kruskal-Wallis and post hoc Dunn’s test. ns: *p* > 0.05, *: *p* ≤ 0.05, **: *p* ≤ 0.01, ***: *p* ≤ 0.001, ****: *p* ≤ 0.0001. Contour of the scatter plot of the height of actin edge and the gradient angle along actin edge. The colormap indicates the density. Histogram of the gradient angle along the actin edge and histogram of the height of actin edge. (**c**) Untreated cells. (**d**) Rho-activator treated cells. (**e**) Blebbistatin-treated cells. (**f**) Y27632 treated cells. The white cross in Plot (**c**–**f**) shows the maximum of the untreated cell.

**Figure 6 cells-11-00430-f006:**
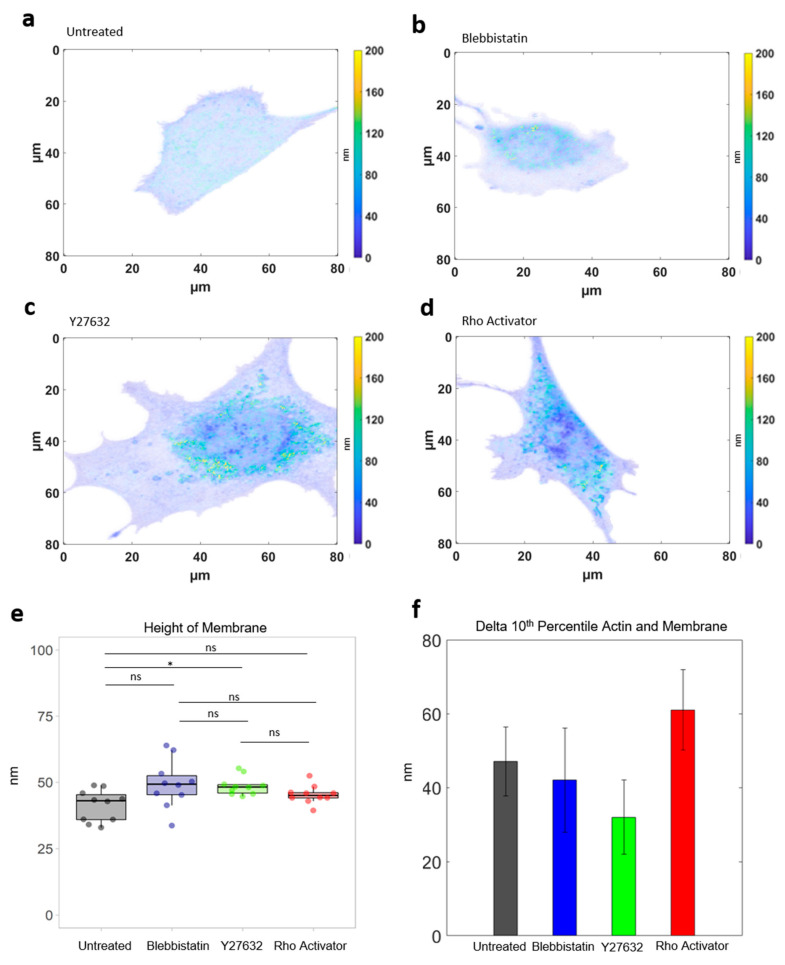
Cell membrane analysis. Intensity weighted height image of the cell membrane (**a**) Untreated cell. (**b**) Blebbistatin-treated cells. (**c**) Y27632-treated cells. (**d**) Rho-activator-treated cells. (**e**) Height of membrane. Test: Kruskal-Wallis and post hoc Dunn’s test. ns: *p* > 0.05, *: *p* ≤ 0.05. (**f**) Difference between 10th Percentile actin and membrane.

**Figure 7 cells-11-00430-f007:**
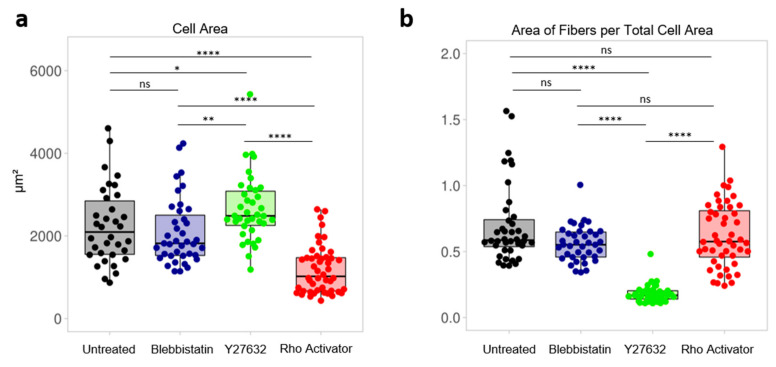
Focal plane analysis. Each dot represents the median of a single cell. (**a**) Cell area of untreated and blebbistatin-, Y27632-, Rho activator-treated cells. (**b**) Area of actin fibers per total cell area of untreated cells and cells treated with blebbistatin, Y27632 and Rho activator. Test: Kruskal-Wallis and post hoc Dunn’s test. ns: *p* > 0.05, *: *p* ≤ 0.05, **: *p* ≤ 0.01, ****: *p* ≤ 0.0001.

**Figure 8 cells-11-00430-f008:**
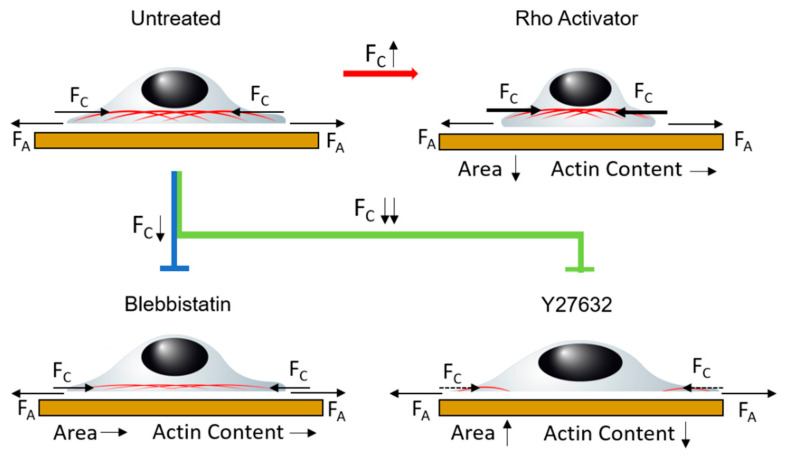
Application of the model of equilibrium between adhesion tension F_A_ and cytoskeleton tension F_C_ in untreated cells. Treatment with blebbistatin and Y27632 as inhibitors and Rho activator as promoter changes the cytoskeleton tension and leads to an imbalance. This changes the height of the actin cytoskeleton and the morphology.

**Table 1 cells-11-00430-t001:** Summary of the medians ± std from height and angle analyses of the actin cytoskeleton and membrane of untreated cells and cells treated with blebbistatin, Y27632 and Rho activator.

Median	Untreated	Blebbistatin	Y27632	Rho Activator
Height of Actin (nm)	114 ± 12	110 ± 17	97 ± 15	124 ± 12
Stress Fiber Height (nm)	118 ± 12	122 ± 17	112 ± 13	132 ± 10
Relative Stress Fiber Rise	0.38 ± 0.04	0.36 ± 0.05	0.42 ± 0.05	0.32 ± 0.05
Height of Actin Edge (nm)	116 ± 13	114 ± 18	103 ± 15	130 ± 12
Gradient Angle along Actin Edge (°)	3.3 ± 1.1	3.0 ± 0.5	3.0 ± 0.8	3.4 ± 0.5
Height of Membrane (nm)	43 ± 6	49 ± 9	48 ± 4	45 ± 4

## Data Availability

Data available on request from the authors.

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
