# Peer review of "Influence of ROCK Pathway Manipulation on the Actin Cytoskeleton Height"

_cells, 2022, doi:10.3390/cells11030430_

Round 1

Reviewer 1 Report

The authors described an interesting method,which to measure the actin height to the substrate and the gradient of the actin height along the edge of the cell. Their results provide an effective method to further observe the cytoskeleton tension. 
However, there are one suggestion: Wilcoxon-rank-sum test is normally used in the comparison of two groups. Kruskal-Wallis test is suitable for comparison among multiple groups.

Author Response

First we would like to thank the reviewers for their time and effort to help us improving the manuscript. The changes we made to the manuscript are highlighted in yellow. Please find a point by point response below.

Reviewer 1:

Dear Reviewer 1,

we would like to thank you for your reading of our manuscript and your constructive suggestions for improving it. We have now replaced the Wilcoxon-rank sum test by the Kruskal Wallis test with Post Hoc Dunn's test.

Reviewer 2 Report

This work aims at investigating actin dynamics through the manipulation of the ROCK pathway. The authors provide a three-dimensional readout of the cytoskeleton using MIET (metal-induced energy transfer). This work represents an innovative approach to analyzing actin dynamics, thus providing a powerful tool for studies providing novel biological mechanisms in health and disease.  
Overall, the manuscript is well written with great experimental design and comprehensive methods description. 
I suggest accepting this manuscript in the present form.

Author Response

First we would like to thank the reviewers for their time and effort to help us improving the manuscript. The changes we made to the manuscript are highlighted in yellow. Please find a point by point response below. 

Thank you for supporting our manuscript!

Reviewer 3 Report

In the manuscript entitled “Influence of ROCK Pathway Manipulation on the Actin Cytoskeleton Height”, Grandy et al. used metal-induced energy transfer (MIET) as a tool to quantitatively examine the third dimension of the actin cytoskeleton with nanometer accuracy, particularly examining the influence of different cytoskeletal drugs on the actin organization. Overall, I believe the manuscript needs several modifications that precludes publication in its current format. My comments are listed below:

1- In the introduction, the authors mention that MIET is a “nanometric precision” tool (line 79) to obtain “super-resolution in the Z dimension” (line 89). It would be interesting to mention this precision in terms of numbers. What is the resolution limit?

2- In addition, the authors mention that they use MIET to quantitatively observe the height of stress fibers and actin cortex (lines 96-97). At first glance, this would be an extremely interesting measurement, particularly for the actin cortex. However, they seem to confuse the actin cortex with the cell outline in a 2D image. Their measurements are performed at this cell outline and not in the actin cortex. This needs to be clearer in the text to avoid misunderstandings.

3- Lines 211-212: the letters a, b, c and d do not correspond to figures 2a, 2b, 2c and 2d. Indeed, it is quite confusing. I suggest renaming them with another code.

4- The authors say that they measure the height distribution of all actin components inside the cell. Does it include actin above and around the nucleus, such as the perinuclear actin cap? If not, it is recommended to change the name.

5- In reference 31, Chizhik et al used MIET to determine the distance between the substrate and the lipid bilayer of cells. They found numbers ranging between 30 and 70nm for different cell types. This distance was never mentioned in the present manuscript. Knowing that actin is above the lipid bilayer, how could these previously found numbers influence the results of the present manuscript? It would be interesting to perform dual stainings for the lipid bilayer and actin, in order to know, for example, if the bilayer is getting closer to the substrate or moving further away according to the treatments used.

6- What novelties does this manuscript bring to the cell biology and to the cytoskeleton fields? The authors need to make this clearer in the manuscript. Otherwise, it seems like a more accurate tool for measuring cytoskeleton height, which I think is totally valid but I would not consider Cells MDPI as the leading journal to publish the results.

Author Response

First we would like to thank the reviewers for their time and effort to help us improving the manuscript. The changes we made to the manuscript are highlighted in yellow. Please find a point by point response below.

Dear Reviewer 3,

we would like to thank you for your critical reading of our manuscript and your constructive suggestions for improving it. We tried to answer all your concerns and think that with your help we managed to significantly improve the manuscript.

Comment 1:

„1- In the introduction, the authors mention that MIET is a “nanometric precision” tool (line 79) to obtain “super-resolution in the Z dimension” (line 89). It would be interesting to mention this precision in terms of numbers. What is the resolution limit?“

Thanks for bringing this to our attention. The axial resolution limit was described by Chizhik et al in reference 31 and is up to 3 nm. This has now been added in line 80 of the manuscript.

Comment 2:

„2- In addition, the authors mention that they use MIET to quantitatively observe the height of stress fibers and actin cortex (lines 96-97). At first glance, this would be an extremely interesting measurement, particularly for the actin cortex. However, they seem to confuse the actin cortex with the cell outline in a 2D image. Their measurements are performed at this cell outline and not in the actin cortex. This needs to be clearer in the text to avoid misunderstandings.“

The edge is defined by the actin staining. Since the actin cortex is at the edge of the cell, the actin perimeter reflects the cortex perimeter. To avoid edge effects, the cell was eroded by 2 pixels as described in the method section. But you are certainly right that this is not a direct measurement of the cortex. Therefore, we worded more cautiously. (line 97 ff, line 225, line 302 ff

Comment 3:

„3- Lines 211-212: the letters a, b, c and d do not correspond to figures 2a, 2b, 2c and 2d. Indeed, it is quite confusing. I suggest renaming them with another code.“

We have now omitted the letter numbering to avoid being confused by the figure numbering.

Comment 4:

„4- The authors say that they measure the height distribution of all actin components inside the cell. Does it include actin above and around the nucleus, such as the perinuclear actin cap? If not, it is recommended to change the name.“

That is absolutely true. We only measure the lower actin layer in the focal volume of our confocal and not the total actin content. This is also not possible with MIET. We have now tried to make this more clear and renamed it with “all basal actin components“.

Comment 5:

„5- In reference 31, Chizhik et al used MIET to determine the distance between the substrate and the lipid bilayer of cells. They found numbers ranging between 30 and 70nm for different cell types. This distance was never mentioned in the present manuscript. Knowing that actin is above the lipid bilayer, how could these previously found numbers influence the results of the present manuscript? It would be interesting to perform dual stainings for the lipid bilayer and actin, in order to know, for example, if the bilayer is getting closer to the substrate or moving further away according to the treatments used.“

Thank you for this suggestion. Unfortunately, it is not possible to perform measurements of two stainings at the same time on our setup. Therefore, as described by Chizhik et al in reference 31, only measurements of the membrane under the conditions were now performed and then averaged. A completely new figure was inserted into the manuscript. This includes images of the cell membrane, a plot of the membrane height and a comparison to the 10th percentile of actin fibers. It shows that the membrane height does not differ under the conditions. These findings are very significant for our manuscript.

Comment 6:

„6- What novelties does this manuscript bring to the cell biology and to the cytoskeleton fields? The authors need to make this clearer in the manuscript. Otherwise, it seems like a more accurate tool for measuring cytoskeleton height, which I think is totally valid but I would not consider Cells MDPI as the leading journal to publish the results.“

Thanks for pointing out that we need to better describe the impact of this work. In fact, your suggestion to measure the basal membrane helps immensely in pinpointing the impact. The distance between actin and membrane is directly related to the remodulation of the connectors between the two structures, the focal adhesions. Therefore, our measurements are the first experimental results showing that by balance between cytoskeletal tension and adhesion tension, increase in tensions stretches the focal adhesions. We discussed this more thoroughly on line 413 ff.

Round 2

Reviewer 3 Report

The authors have adressed all comments raised by this reviewer.

I would only suggest to remove the sentence "the cortex is larger than the here described structures" - line 303.

Apart from that, I suggest publishing the manuscript.

Author Response

Thanks for reviewing so quickly!! We deleted the sentence.